# Identifying Factors Associated with the Efficacy of Lasmiditan 50 mg as an Acute Treatment for Migraine Attacks Under Various Dosing Conditions in Real-World Clinical Practice

**DOI:** 10.3390/neurolint17050062

**Published:** 2025-04-22

**Authors:** Takafumi Tanei, Shun Yamamoto, Satoshi Maesawa, Yusuke Nishimura, Tomotaka Ishizaki, Yoshitaka Nagashima, Yoshiki Ito, Miki Hashida, Takahiro Suzuki, Hajime Hamasaki, Toshihiko Wakabayashi, Ryuta Saito

**Affiliations:** 1Department of Neurosurgery, Graduate School of Medicine, Nagoya University, Nagoya 466-8550, Japanmaesawa.satoshi.n0@a.mail.nagoya-u.ac.jp (S.M.); yusukenishimura0411@gmail.com (Y.N.); ishizaki.tomotaka.d8@f.mail.nagoya-u.ac.jp (T.I.); nagashima4251@gmail.com (Y.N.); yorimichi33@gmail.com (Y.I.); licorice8823@gmail.com (M.H.); hywtts.tetu@gmail.com (T.S.); marukoman0156@gmail.com (H.H.); saito.ryuta.b1@f.mail.nagoya-u.ac.jp (R.S.); 2Department of Specialized Headache Outpatient, Nagoya Garden Clinic, Nagoya 451-0051, Japan; wakaba@garden-cl.jp; 3Department of Neurosurgery, National Hospital Organization, Nagoya Medical Center, Nagoya 460-0001, Japan

**Keywords:** lasmiditan, migraine, 50 mg

## Abstract

**Background/Objectives**: Lasmiditan is a newly developed drug for the acute treatment of migraine attacks, but factors associated with its efficacy remain unclear. This study aimed to confirm the efficacy of lasmiditan started at 50 mg under various dosing conditions and identify factors associated with its efficacy. **Methods**: There are four reasons for prescribing lasmiditan: as an add-on to triptan, if triptan is ineffective, if triptan produces side effects, and when triptan is contraindicated. Lasmiditan was administered at a dose of 50 mg. The efficacy of lasmiditan was defined as the disappearance of headache or a 50% or greater reduction in headache intensity within two hours after dosing. This study included 108 patients with migraines who took lasmiditan. **Results**: The results for efficacy and the side effects of lasmiditan were as follows: effective without side effects (22), effective with mild side effects (32), ineffective (14), and severe side effects (40). The efficacy rate of lasmiditan 50 mg was 50.0% (54/108). The following factors were found to be associated with lasmiditan’s efficacy: sex, migraine classification, calcium channel blockers, and anti-calcitonin gene-related peptide monoclonal antibody (CGRP-mAb) treatment. The overall incidence of side effects was 66.7%, and the dropout rate was 37.0%. Somnolence was more prevalent in the effective group, and other side effects were more prevalent in patients who dropped out due to the side effects of lasmiditan. **Conclusions**: Lasmiditan is likely to be effective in males with severe migraine classification and receiving CGRP-mAb treatment. If mild somnolence is a side effect, the drug can be continued and may be effective.

## 1. Introduction

Migraine is a common disabling neurological disorder that is estimated to affect more than one billion people worldwide [1,2]. Migraine attacks can be accompanied by severe headaches, vomiting, and hypersensitivity to various stimuli, reducing an individual’s quality of life [2]. Furthermore, the cost–productivity loss associated with presenteeism due to migraine was estimated at 21.3 billion US$/year in Japan as a whole [3]. Therefore, the appropriate treatment of migraine is important not only for improving the individual’s quality of life, but also from a national economic perspective.

Migraine treatments are divided into acute medications that suppress migraine attacks and prophylactic medications that reduce the frequency and intensity of headaches [2]. Acute medications for migraine attacks mainly involve the use of triptans, which have vasoconstrictive effects. Nonsteroidal anti-inflammatory drugs (NSAIDs) and over-the-counter drugs are also used to treat acute migraine attacks, but their effectiveness is limited. Oral prophylactic medications for migraine mainly include anticonvulsants, antidepressants, calcium (Ca) channel blockers, and beta-blockers. In recent years, prophylactic treatments have changed dramatically with the introduction of anti-calcitonin gene-related peptide monoclonal antibody (CGRP-mAb) treatment. CGRP-mAb treatment provided patients with a significant reduction in headache frequency and intensity [4,5,6,7,8,9]. However, even the administration of these highly effective CGRP-mAbs cannot completely eliminate migraine attacks, so acute medications for migraine attacks remain important.

Triptans, and 5-HT_1B_ and 5-HT_1D_ receptor agonists, are the mainstay of acute medications for migraine attacks, but they do have some problems. First, triptans may be ineffective or may not be usable due to side effects. Second, triptans need to be taken as early as possible after the onset of a migraine attack; the late taking of triptans has been shown to be ineffective [10,11]. Third, even if triptans are effective, headaches may recur as they lose effectiveness over time. Finally, triptans have vasoconstrictive effects and are therefore contraindicated for some patients, such as those diagnosed with cardiovascular disease, cerebrovascular disease, hemiplegic migraine, and migraine with brainstem aura.

Lasmiditan is a selective 5-HT_1F_ serotonin receptor agonist recently developed for the acute treatment of migraine attacks [12,13]. It has also been suggested that lasmiditan may act as a partial agonist on 5-HT_1B/D_ receptors and exert anti-migraine effects [14]. Lasmiditan was approved based on the results of the SMURAI trial in the USA and the SPARTAN trail in the EU, and its efficacy in Japanese patients was also demonstrated in the MONONOFU trial [15,16,17]. A recent systematic review and network meta-analysis showed that lasmiditan was effective for the acute treatment of migraine in triptan-insufficient responders [18]. However, significant caution is required regarding the use of lasmiditan due to one of its unique characteristics—its ability to cross central nervous system barriers. In a simulated driving study in healthy subjects, lasmiditan significantly impaired driving ability. Therefore, patients taking lasmiditan should be advised not to drive or engage in other activities requiring heightened attention until at least eight hours after taking each dose of lasmiditan [19].

Compared to triptans, lasmiditan has the advantages of not having a vasoconstrictive effect and being effective even more than two hours after the onset of a migraine [17,20]. However, lasmiditan causes a relatively high rate of dose-dependent side effects such as dizziness, somnolence, malaise, and nausea [17,20,21]. Predictors of side effects with lasmiditan have been reported to include high dosages, non-Hispanic ethnicity, lower patient body mass index (BMI), and the migraine attack treated being of mild or moderate severity [22]. Japanese patients, who had a lower mean BMI, had a higher incidence of side effects from lasmiditan than non-Asian patients, suggesting that starting lasmiditan at 50 mg may reduce the incidence of side effects [17,23,24]. Although there are a few reports on the efficacy and side effects of lasmiditan in real-world practice, factors associated with efficacy have not yet been identified [23,24].

In clinical practice, lasmiditan is prescribed under various conditions, including in combination with a triptan or as lasmiditan alone when triptans are ineffective or contraindicated [18,23,24]. The aim of this study was to confirm the efficacy of lasmiditan started at 50 mg under various dosing conditions in real-world clinical practice, as well as to identify factors associated with its efficacy. This is an open study without blind randomization.

## 2. Materials and Methods

### 2.1. Study Design

This was a single-center, retrospective, real-world study of migraine patients prescribed lasmiditan. The patients were recruited from the specialized headache outpatient clinic at the Nagoya Garden Clinic from May 2022 to December 2024. All patients had a diagnosis of migraine according to the International Classification of Headache Disorders 3 criteria [25]. Patients first underwent magnetic resonance imaging to exclude intracranial diseases, and then medical treatment for migraine was started. Diagnosis and treatment were performed by a neurosurgeon specializing in the field of pain and headaches (T.T.). Lasmiditan was prescribed to migraine patients in the following cases: when triptan was effective but headache recurred when the effect wore off; when two or more triptans had been tried without effect; when triptans could not be used due to side effects; and when triptans were contraindicated. The initial dose of lasmiditan was 50 mg. If it was effective, the dose was not increased. If there were neither side effects nor efficacy and the patient wished, the dose was increased to 100 mg. Patients eligible for the outcome analysis met the following criteria: being lasmiditan treatment-naïve, actually taking lasmiditan, and subsequently attending the outpatient clinic to evaluate the efficacy of lasmiditan. This study was approved by the Ethics Review Committee of Nagoya University Graduate School of Medicine (approval number 2022-0316). Since this study was noninvasive, the Ethics Review Committee of Nagoya University Graduate School of Medicine waived the requirement for written, informed consent from the patients, but the opt-out method was adopted in accordance with Japanese ethics guidelines. This research was completed in accordance with the Declaration of Helsinki as revised in 2013.

### 2.2. Data Collection

Demographic data (age, sex, onset years of migraine, family history of headache, history of psychiatric disorders, migraine with aura, classification of migraine, oral prophylactic medications available, and use of CGRP-mAb injection) were collected retrospectively. Migraines were classified into four types: episodic migraine (EM), high-frequency episodic migraine (HFEM), chronic migraine (CM), and medication overuse headache (MOH). Monthly headache days of 0–7 days were defined as EM, and 8–14 days as HFEM. Most patients recorded headaches, migraines, and the number of acute medication intakes in their headache diaries. Lasmiditan, triptan, and non-triptan medications (e.g., NSAIDs, acetaminophen, and over-the-counter drugs) were considered acute medications. The choice of oral prophylactic medicines was based on the experience of the treating physician. When patients agreed to start one of the three types of CGRP-mAb treatment, the choice of medication was determined through discussions between the patient and the physician in the clinical setting. Data on lasmiditan were collected, including reasons for prescribing, efficacy, and side effects. The reasons for prescribing lasmiditan were collected in the following four ways: as an add-on to triptan, if triptan was ineffective, if triptan produced side effects, and if triptan was contraindicated. Lasmiditan was judged to be effective if the headache disappeared or the headache intensity was reduced by 50% or more within two hours after taking the drug. The side effects of lasmiditan, such as dizziness, somnolence, malaise, nausea, and others, were collected. Side effects were defined as severe if they were so severe that patients could not continue taking lasmiditan, and mild if patients could continue taking lasmiditan despite the side effects.

### 2.3. Assessments and Statistical Analysis

The primary endpoints of the study were the efficacy and side effect rates when treatment was started with lasmiditan 50 mg. Based on the efficacy and side effects of lasmiditan, patients were classified into the effective group (effective without side effects, and effective with mild side effects) and the ineffective group (ineffective and severe side effects). The two groups were compared to identify factors associated with the efficacy of lasmiditan. In patients who experienced side effects from lasmiditan, the breakdown of side effects was compared between patients in whom lasmiditan was effective with mild side effects and patients who experienced severe side effects. For continuous variables, the Wilcoxon signed-rank sum test was performed. For categorical variables, when the expected frequency in any category was less than 5, Fisher’s exact test was applied, and when all expected frequencies were 5 or greater across all categories, the chi-squared test was applied. Significance was set at *p* < 0.05. The statistical analyses were carried out using R version 4.3.2 (R Foundation for Statistical Computing, Vienna, Austria) and RStudio (version 2023.12.0; RStudio, Inc., 2022, Boston, MA, USA).

## 3. Results

### 3.1. Participants’ Demographic Characteristics

From May 2022 to December 2024, 1401 new patients with headache symptoms visited the specialized headache outpatient clinic, and 960 of them (68.5%) were diagnosed with migraines. The numbers of migraine types were as follows: EM 566, HFEM 180, CM 111, and migraine with MOH 103. Of the 960 patients with migraines, 140 were newly prescribed lasmiditan 50 mg, and 108 of them actually took the medication (Figure 1).

The clinical characteristics of the eligible patients are shown in Table 1. The mean age was 34.6 ± 12.3 years, with females accounting for 83.3% of eligible patients (90/108). The onset ages of migraine were teenage years and younger (*n* = 65, 60.2%), 20s (*n* = 31, 28.7%), 30s (*n* = 9, 8.3%), and 40s and older (*n* = 3, 2.8%). Seventy-two patients had a family history of headaches (66.7%), and thirteen patients had a history of psychiatric disorders (12.0%), with depression in eight (7.4%) and others in five (4.6%). The migraine type was migraine with aura in 49 patients (45.4%), and the classification of migraine included EM (*n* = 57, 52.8%), HFEM (*n* = 20, 18.5%), CM (*n* = 16, 14.8%), and migraine with MOH (*n* = 15, 13.9%). Eighty patients (74.1%) were taking oral prophylactic medications, of whom forty (42.6%) were taking a single medication, and thirty-four (31.5%) were taking multiple medications. The types of oral prophylactic medications included anticonvulsants (63), antidepressants (41), Ca channel blockers (23), and beta-blockers (2). Thirty-seven patients were receiving CGRP-mAb treatment (34.3%).

### 3.2. Efficacy of Lasmiditan

Of the 108 patients enrolled, lasmiditan was effective without any side effects in 22, effective but had mild side effects in 32, neither effective nor had any side effects in 14, and had severe side effects in 40 (Figure 2). Therefore, 54 patients were classified into the effective group and 54 into the ineffective group, with an efficacy rate of lasmiditan 50 mg of 50.0% (54/108). The results of the comparison between the effective and ineffective groups are shown in Table 2. There were no significant differences between the two groups in age, onset age, family history of headache, history of psychiatric disorders, migraine with aura, or oral prophylactic medication available. The most common reason for prescribing lasmiditan was as an add-on to triptan, and no differences were noted in the prescribing conditions. Significant differences were observed between the two groups in sex (*p* = 0.039), EM (*p* = 0.012), CM (*p* = 0.03), Ca channel blockers (*p* = 0.034), and CGRP-mAb treatment (*p* = 0.0008). In one of the fourteen patients who had neither side effects nor benefit from lasmiditan 50 mg, lasmiditan 100 mg was tried, but no benefit was observed. Of the fifty-four patients who responded to lasmiditan 50 mg, six patients tried lasmiditan 100 mg, and in one patient, the dose was decreased to 50 mg due to side effects. The remaining 48 patients continued taking the drug as needed without increasing the dose.

### 3.3. Side Effects of Lasmiditan

Side effects of lasmiditan 50 mg occurred in 72 patients, with an incidence rate of 66.7% (72/108). Of these, 40 patients (37.0%) discontinued lasmiditan 50 mg due to severe side effects. Of the 72 patients who experienced side effects, 20 (27.8%) experienced several different side effects. The breakdown of side effects (with overlap) was dizziness (28), somnolence (33), malaise (10), nausea (7), and others (15). Other symptoms included the following: insomnia, palpitations, cold sweats, weakness, a feeling of floating, trembling hands, numbness in hands and feet, and strange dreams. The results of the comparison of the types of side effects between the 32 patients in whom treatment was effective with mild side effects and the 40 patients who experienced severe side effects are shown in Table 3. Significant differences between the two groups were observed in somnolence (*p* = 0.039) and others (*p* = 0.0008).

## 4. Discussion

The present study showed that 108 migraine patients actually took lasmiditan 50 mg as acute treatment for migraine attacks, and that it was effective in 54 of them, with an efficacy rate of 50.0%. Factors associated with lasmiditan efficacy were sex, migraine classification, Ca channel blockers, and CGRP-mAb treatment. The overall incidence of side effects with lasmiditan 50 mg was 66.7%, and the dropout rate due to severe side effects was 37.0%. As regards the side effects of lasmiditan, somnolence was more prevalent in the effective group, and other side effects were more prevalent in patients who dropped out due to the side effects of lasmiditan.

In real-world clinical practice, there are two reports regarding the efficacy and side effects of lasmiditan in Japanese patients [23,24]. Shibata et al. reported the efficacy of the combination of triptan and lasmiditan for the treatment of migraine [23]. In their study, a triptan was first administered as an acute treatment for a migraine attack, and if headache relief was less than 50%, lasmiditan 50 mg was administered within two hours of the onset of the migraine. The analysis of 20 patients and 40 migraine attacks showed that pain was reduced in 32 migraine attacks (80%) after the addition of lasmiditan. Adverse events such as dizziness and somnolence were observed in 25 migraine attacks (63%), but most were mild and resolved quickly. Ishii et al. reported the efficacy of lasmiditan 100 mg taken alone for the treatment of migraine attacks [24]. Twenty-three of forty-eight patients (47.9%) experienced two headache-free hours after taking 100 mg of lasmiditan, and twenty (41.7%) patients preferred lasmiditan to their previous acute medication. However, the incidence of side effects was high, at 91.7%, occurring in 44 patients. The unique feature of the present study that differs from the above two studies was that it showed the efficacy and side effects of lasmiditan in patients receiving various treatments rather than in patients with a uniform prescription condition. The results of the present study were obtained in real-world clinical practice, and will provide useful information to clinicians involved in the treatment of migraine. The efficacy of lasmiditan cannot be simply compared between these reports and the present study because there were differences in patient background, initial dose, administration conditions, and efficacy evaluation methods. However, in ethnic groups with a lower mean BMI, the incidence of side effects with lasmiditan is higher at 100 mg than at 50 mg, so an initial dose of 50 mg may be preferable to reduce side effects.

This is the first report to identify factors associated with lasmiditan efficacy, including sex, migraine classification, Ca channel blockers, and CGRP-mAb treatment. First, the proportion of males was significantly higher in the effective group. This result suggests that, because the occurrence of side effects from lasmiditan is associated with lower BMI [22], males, who generally have a higher BMI than females, benefited from lasmiditan without dropping out due to side effects. Second, in the classification of migraine, the effective group had significantly fewer EM and significantly more CM cases. This suggests that patients classified as having severe migraines may respond better to lasmiditan or be less likely to discontinue treatment due to side effects. One predictor of side effects with lasmiditan is migraine attacks that are mild or moderate in severity [22]. Conversely, those with severe migraine attacks are less likely to develop side effects and are more likely to benefit from lasmiditan. It is thought that lasmiditan was primarily used to treat severe migraine attacks in patients with CM. In addition, patients with CM may have used several acute and prophylactic medications in the past and may therefore be more tolerant to the side effects of these medications. Third, one of the reasons why the number of patients receiving CGRP-mAb treatment was significantly higher in the effective group may be that the number of patients with CM and MOH, who are in high need of CGRP-mAb treatment [4,5,6,7,8,9], was twice as high as that in the ineffective group. Incidentally, the improved efficacy of triptans has been reported with CGRP-mAb treatment, implying that CGRP-mAb treatment may also improve the efficacy of other acute medications [26]. Although the following opinion is speculation not supported by experimental information, in the present study, the significantly higher number of patients receiving CGRP-mAb treatment in the efficacy group may not only have been due to the higher number of patients with severe migraines, but may also be the first data to show that CGRP-mAb treatment enhanced the efficacy of lasmiditan. Finally, no significant differences were seen with other oral prophylactic medications, whereas only Ca channel blockers showed significant differences, although the mechanism of this result is unclear.

The incidence of all side effects from lasmiditan 50 mg in the present study was 66.7%, which was almost the same as the 63% reported previously when the same dose was taken [23]. The breakdown of side effects was consistent with previous reports, with dizziness and somnolence being the most common [15,16,17,23,24]. New findings regarding the side effects of lasmiditan in the present study were that somnolence was more prevalent in the effective group, and other side effects were more prevalent in patients who dropped out due to the side effects of lasmiditan. The results suggest that the mild side effect of somnolence can be tolerated with a short break or sleep, resulting in the beneficial use of lasmiditan. Therefore, patients who experience the mild side effect of somnolence can benefit from lasmiditan if they are instructed to take it on days off or in the evening, when conditions are favorable for sleep. Other side effects, such as insomnia, palpitations, cold sweats, weakness, trembling, and numbness were rare but specific symptoms that caused significant discomfort and prevented patients from continuing to take lasmiditan. Having discussed the efficacy, efficacy-associated factors, and side effects of lasmiditan, it has been noted that, for the latest approved drugs for the management of acute migraine, such as lasmiditan, ubrogepant, and rimegepant, it remains difficult to justify their use except in special populations/conditions due to the lack of clinical trials investigating the consistency of long-term safety and efficacy [27].

The present study has limitations that should be noted. First, this was a single-center, retrospective, small case series, which may lead to unstable results, and it did not demonstrate the effectiveness of lasmiditan compared with a control group. Second, criteria were not set for the timing of starting lasmiditan treatment. Third, the number of cases in which the dose of lasmiditan was increased from 50 mg to 100 mg was small, so the efficacy of lasmiditan may have been underestimated. Fourth, the efficacy of lasmiditan was judged by a 50% or greater reduction in headache intensity within two hours of taking the drug, and therefore did not distinguish between headache relief and the disappearance of headache. Fifth, the present study did not collect information about patients’ height and weight, so the results could not be corroborated with actual BMI data. To clarify the exact relationship between BMI and the efficacy and side effects of lasmiditan, these data should be collected in future studies. Sixth, because this study was a retrospective analysis, confounding factors such as concomitant medications and patient compliance may have influenced the results. Seventh, the long-term outcomes of lasmiditan are still unknown. Finally, because this was a non-randomized drug trial, the results of this study alone are not sufficient to prove the usefulness of the compound.

## 5. Conclusions

The present study showed that the efficacy rate of lasmiditan started at 50 mg under various dosing conditions was 50.0%, and that factors associated with efficacy included sex, migraine classification, Ca channel blockers, and CGRP-mAb treatment. The overall incidence of the side effects of lasmiditan 50 mg was 66.7%, with a dropout rate of 37.0%. We suggest medication strategies for the use of lasmiditan in East Asians. The initial dose should be 50 mg and the severity of side effects should be monitored. Patients who may benefit from lasmiditan include males, patients with CM, and patients receiving CGRP-mAb treatment.

## Figures and Tables

**Figure 1 neurolint-17-00062-f001:**
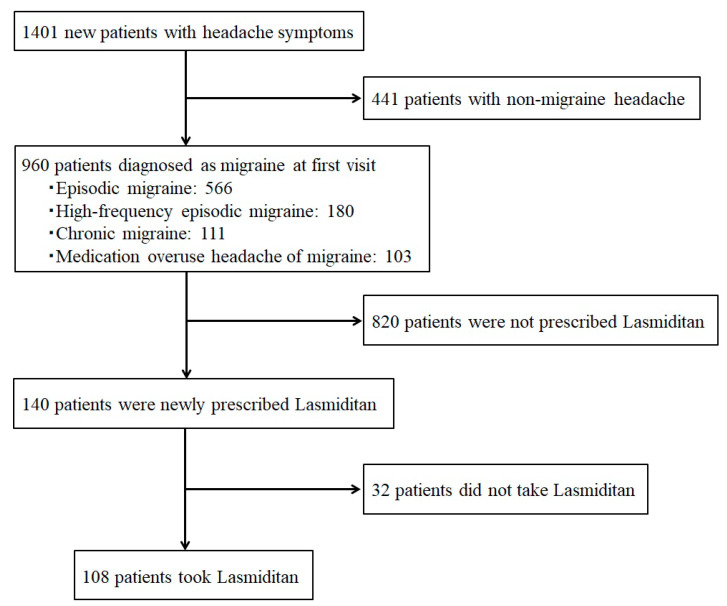
Flowchart showing patient selection.

**Figure 2 neurolint-17-00062-f002:**
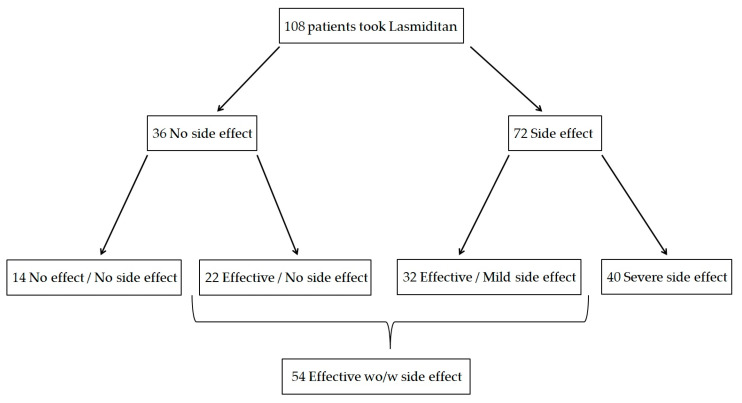
Flowchart showing efficacy and side effects of lasmiditan. w: with, wo: without.

**Table 1 neurolint-17-00062-t001:** Demographic and clinical characteristics of the patients.

Characteristics	*n* = 108
Age (years), mean ± SD	34.6 ± 12.3
Sex, female; *n* (%)	90 (83.3%)
Onset years; *n* (%)	
Teens and younger	65 (60.2%)
20s	31 (28.7%)
30s	9 (8.3%)
40s and older	3 (2.8%)
Family history of headaches; *n* (%)	72 (66.7%)
History of psychiatric disorders; *n* (%)	13 (12.0%)
Depression	8 (7.4%)
Others	5 (4.6%)
Migraine with aura; *n* (%)	49 (45.4%)
Classification of migraine	
EM	57 (52.8%)
HFEM	20 (18.5%)
CM	16 (14.8%)
Migraine with MOH	15 (13.9%)
Oral prophylactic medication available	80 (74.1%)
Single	46 (42.6%)
Multiple	34 (31.5%)
Types of prophylactic medications (includes duplicates)	
Anticonvulsants	63 (58.3%)
Antidepressants	41 (38.0%)
Ca channel blockers	23 (21.3%)
Beta-blockers	2 (1.9%)
CGRP-mAb treatment	37 (34.3%)

Ca: calcium, CGRP-mAb: anti-calcitonin gene-related peptide monoclonal antibody, CM: chronic migraine, EM: episodic migraine, HFEM: high-frequency episodic migraine, MOH: medication overuse headache, *n*: number, SD: standard deviation.

**Table 2 neurolint-17-00062-t002:** Comparison of effective and ineffective groups.

	Effective Group	Ineffective Group	*p*-Value
	*n* = 54	*n* = 54	
Age (years), mean ± SD	33.7 ± 10.6	35.5 ± 13.8	0.5
Sex, female; *n* (%)	41 (75.9%)	49 (90.7%)	0.039
Onset years; *n* (%)			
Teens and younger	33 (61.1%)	32 (59.3%)	0.8
20s	15 (27.8%)	16 (29.6%)	0.8
30s	4 (7.4%)	5 (9.3%)	>0.9
40s and older	2 (3.7%)	1 (1.9%)	>0.9
Family history of headaches; *n* (%)	39 (72.2%)	33 (61.1%)	0.2
History of psychiatric disorders; *n* (%)	6 (11.1%)	7 (13.0%)	0.8
Migraine with aura; *n* (%)	23 (42.6%)	26 (48.1%)	0.6
Classification of migraine			
EM	22 (40.7%)	35 (64.8%)	0.012
HFEM	11 (20.4%)	9 (16.7%)	0.6
CM	12 (22.2%)	4 (7.4%)	0.03
Migraine with MOH	9 (16.7%)	6 (11.1%)	0.4
Oral prophylactic medication available	42 (77.8%)	38 (70.4%)	0.4
Types of prophylactic medications (includes duplicates)			
Anticonvulsants	32 (59.3%)	31 (57.4%)	0.8
Antidepressants	22 (40.7%)	19 (35.2%)	0.6
Ca channel blockers	16 (29.6%)	7 (13.0%)	0.034
Beta-blockers	2 (3.7%)	0 (0.0%)	0.5
CGRP-mAb treatment	25 (46.3%)	12 (22.2%)	0.0008
Reasons for prescribing lasmiditan			
Add-on to triptan	43 (79.6%)	46 (85.2%)	0.6
Triptan ineffective	2 (3.7%)	2 (3.7%)	>0.9
Triptan side effect	4 (7.4%)	1 (1.9%)	0.6
Contraindicated triptan	5 (9.3%)	5 (9.3%)	>0.9

Ca: calcium, CGRP-mAb: anti-calcitonin gene-related peptide monoclonal antibody, CM: chronic migraine, EM: episodic migraine, HFEM: high-frequency episodic migraine, MOH: medication overuse headache, *n*: number, SD: standard deviation.

**Table 3 neurolint-17-00062-t003:** Comparison of side effects between effective with mild side effects and severe side effects.

	Effective with Mild Side Effects	Severe Side Effects	*p*-Value
	*n* = 32	*n* = 40	
Dizziness	11 (34.4%)	17 (42.5%)	0.5
Somnolence	19 (59.4%)	14 (35.0%)	0.039
Malaise	4 (12.5%)	7 (17.5%)	0.5
Nausea	2 (6.3%)	5 (12.5%)	0.5
Others	3 (9.4%)	12 (30.0%)	0.032

## Data Availability

The datasets used and/or analyzed during the current study are available from the corresponding author on reasonable request.

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
