# Peer review of "Identifying Factors Associated with the Efficacy of Lasmiditan 50 mg as an Acute Treatment for Migraine Attacks Under Various Dosing Conditions in Real-World Clinical Practice"

_2035-8377, 2025, doi:10.3390/neurolint17050062_

Round 1

Reviewer 1 Report

Comments and Suggestions for Authors

Lasmiditan, a selective 5-HT1F receptor agonist, is a drug for the acute treatment of migraine that does not cause vasoconstriction, unlike traditional triptans.

The manuscript addresses a long-standing clinical challenge: the acute treatment of migraine, where clinicians must carefully balance efficacy with an acceptable safety profile. However, in this case, the authors have completely omitted any mention, description, or discussion of a key characteristic of this drug—its ability to cross the CNS barrier and produce significant central effects. They have overlooked the most critical aspect, the one flagged by both the FDA and EMA, which led to the mandatory warning requiring patients to refrain from driving for 8 hours after administration.

Reyvow – Lilly
PMID: 32449213
EMA document

See here: "4.4 Special warnings and precautions for use
Central nervous system (CNS) effects and driving impairment
Lasmiditan is associated with CNS adverse reactions. In a simulated driving study in healthy subjects, lasmiditan significantly impaired the ability to drive (see section 4.7). Patients should be advised not to drive or engage in other activities requiring heightened attention until at least 8 hours after taking each dose of lasmiditan, even if they feel well enough to do so. Patients who cannot follow this advice should not take lasmiditan."

I would recommend including the following recent references to strengthen different sections of the manuscript:

Introduction: The seminal paper on migraine PMID: 39482575

5-HT1F receptor action: PMID: 35177004

Use in insufficient responders: PMID: 39516789

Critical discussion: DOI: 10.1007/s42399-020-00390-1

Author Response

Reviewer #1:

Thank you for your valuable comments and suggestions. We cited the four references you suggested in the text and added them to the Reference. Specifically, we revised the text as follows.

Reyvow – Lilly PMID: 32449213 EMA document

Response: We added the following sentence of regarding lasmiditan's ability to cross the central nervous system barriers to the introduction (page 2, line: 30-35).

“However, significant caution is required regarding the use of lasmiditan due to one of its unique characteristics, its ability to cross central nervous system barriers. In a simulated driving study in healthy subjects, lasmiditan significantly impaired driving ability. Therefore, patients taking lasmiditan should be advised not to drive or engage in other activities requiring heightened attention until at least eight hours after taking each dose of lasmiditan [19].”

Introduction: The seminal paper on migraine PMID: 39482575

Response: We cited your suggested paper as [2] in the introduction to a general discussion of migraine.

5-HT1F receptor action: PMID: 35177004

Response: The following sentence was added to the introduction (page 2, line: 24-36).

“It has also been suggested that lasmiditan may act as a partial agonist on 5-HT1B/D receptors and exert anti-migraine effects [14].”

Use in insufficient responders: PMID: 39516789

Response: The following sentence was added to the introduction (page 2, line: 28-30).

“A recent systematic review and network meta-analysis showed that lasmiditan was effective for acute treatment of migraine in triptan-insufficient responders [18].”

Critical discussion: DOI: 10.1007/s42399-020-00390-1

Response: Sorry, we could not find the suggested paper on PubMed and therefore could not cite it.

Reviewer 2 Report

Comments and Suggestions for Authors

This study provides real-world efficacy and safety data of lasmiditan 50 mg, which fills an evidence gap in East Asian populations.

Here are some comments:

  1. Only 108 patients participated in the study and took lamistatin, among whom male patients accounted for only 16.7% (18/108). The small sample size may lead to unstable results.
  2. While the authors identified "sex, migraine subtype, calcium channel blockers, and CGRP-mAb treatment" as efficacy-related factors, concise mechanistic explanations are needed. For example:How do calcium channel blockers potentially interact with lasmiditan’s mechanism of action?
  3. As a retrospective analysis, it is necessary to supplement the discussion on the limitations caused by confounding factors (such as whether there is combined medication, patient compliance).

  4. As a retrospective analysis, it is necessary to supplement the discussion on the limitations caused by confounding factors (such as whether there is combined medication, patient compliance).

  5. In the discussion section, although the authors cited previous studies, they failed to adequately contrast and explain the uniqueness of this research. For instance, the authors mentioned that the lower BMI of Japanese patients might affect the incidence of side effects, but they did not specifically analyze the BMI data, resulting in the inference lacking direct data support. Additionally, it would be best to supplement the hypothesis that CGRP-mAb enhances efficacy with relevant experimental information to avoid remaining merely at the level of observational association.
  6. In the conclusion section, the author's description is rather general and lacks practicality. As a result, after reading the entire piece, one feels that the story ends abruptly, leaving the reader unsatisfied. It is suggested that the author provide more specific medication strategies for different patient subgroups (such as women and chronic migraine patients), or discuss how to balance efficacy and side effects.

Author Response

Reviewer #2:

Thank you for your valuable comments and suggestions. We responded each of your comments and listed them below.

  1. Only 108 patients participated in the study and took lamistatin, among whom male patients accounted for only 16.7% (18/108). The small sample size may lead to unstable results.

Response: Although we have already stated in the limitations section that the sample size of this study is small, we added a statement that this may make the results unstable (page 9, line: 38).

  1. While the authors identified "sex, migraine subtype, calcium channel blockers, and CGRP-mAb treatment" as efficacy-related factors, concise mechanistic explanations are needed. For example: How do calcium channel blockers potentially interact with lasmiditan’s mechanism of action?

Response: The reasons why “sex”, “migraine subtype”, and “CGRP-mAb treatment” were effect-related factors have already been described in the Discussion (page 8, line: 35- page 9, line: 23), as listed below.

・Sex: Male generally have a higher BMI than female, so they were less likely to drop out due to side effects and therefore benefit from lasmiditan.

・Migraine subtype: Migraine classified as severe were more likely to benefit from lasmiditan for the following reasons: they had a higher number monthly severe headache attacks (making them less likely to experience side effects from lasmiditan), and had previously used a variety of medications, which led to tolerance to lasmiditan.

・CGRP-mAb treatment: The reason why the number of patients who received CGRP-mAb treatment was significantly higher in the effective group may be because the number of patients with CM and MOH, who are in high need of CGRP-mAb treatment, was twice as high as that in the ineffective group.

・Calcium channel blocker: Unlike the three points above, it is unclear how calcium blockers affected the efficacy of lasmiditan. Therefore, we are unable to explain its mechanism of action. To avoid any misunderstanding, the statement in the text that “calcium blockers interacted with the efficacy of lasmiditan” was deleted and revised as follows (page 9, line: 21-23).

Before: Finally, no significant differences were seen with other oral prophylactic medications, whereas only Ca channel blockers showed significant differences, which may have interacted with the efficacy of lasmiditan, although the mechanism is unclear.

After: Finally, no significant differences were seen with other oral prophylactic medications, whereas only Ca channel blockers showed significant differences, although the mechanism of this result is unclear.

  1. As a retrospective analysis, it is necessary to supplement the discussion on the limitations caused by confounding factors (such as whether there is combined medication, patient compliance).
  2. As a retrospective analysis, it is necessary to supplement the discussion on the limitations caused by confounding factors (such as whether there is combined medication, patient compliance).

Response: Comments 3 and 4 are the same sentence, so we only take one response here. The following sentence was added to limitation section (page 9, line: 48-50).

“Sixth, because this study was a retrospective analysis, confounding factors such as concomitant medications and patient compliance may have influenced the results.”

  1. In the discussion section, although the authors cited previous studies, they failed to adequately contrast and explain the uniqueness of this research. For instance, the authors mentioned that the lower BMI of Japanese patients might affect the incidence of side effects, but they did not specifically analyze the BMI data, resulting in the inference lacking direct data support. Additionally, it would be best to supplement the hypothesis that CGRP-mAb enhances efficacy with relevant experimental information to avoid remaining merely at the level of observational association.

Response: Thank you very much for pointing out very important and difficult issues. We respond to each of the three points you mentioned.

・Failed to adequately contrast and explain the uniqueness of this research.

Response: As you pointed out, the characteristics of this research compared to existing studies were not conveyed well. To make it easier for readers to understand, we added the following sentence (page 8, line: 24-29).

“The unique feature of the present study that differs from the above two studies was that it showed the efficacy and side effects of lasmiditan in patients receiving various treatments rather than in patients with a uniform prescription condition. The results of the present study were obtained in a real-world clinical practice, and will provide useful information to clinician involved in the treatment of migraine.”

・They did not specifically analyze the BMI data, resulting in the inference lacking direct data support.

Response: In our specialized headache outpatient clinic, we did not measure the height and weight of patients. Therefore, as you pointed out, we did not have the BMI data of the patients and could not back up the results with data. We already described this point as a problem of this study in the limitations section, but we have explained it in more detail and revised it as follows (page 9, line: 45-48).

Before: Fifth, BMI was not measured, so information on the relationship between lasmiditan efficacy and BMI could not be obtained.

After: Fifth, the present study did not collect information about patients' height and weight, so the results could not be corroborated with actual BMI data. To clarify the exact relationship between BMI and the efficacy and side effects of lasmiditan, these data should be collected in future prospective studies.

・The hypothesis that CGRP-mAb enhances efficacy with relevant experimental information.

Response: We could not find any experimental information to support the hypothesis, so we revised the text as follows to make it clear that the hypothesis is speculation (page 9, line: 17-21).

Before: In the present study, the significantly higher number of patients receiving CGRP-mAb treatment in the efficacy group may not only have been due to the higher number of patients with severe migraine, but may be the first data to show that CGRP-mAb treatment enhanced the efficacy of lasmiditan.

After: Although the following opinion is speculation not supported by experimental information, in the present study, the significantly higher number of patients receiving CGRP-mAb treatment in the efficacy group may not only have been due to the higher number of patients with severe migraine, but may be the first data to show that CGRP-mAb treatment enhanced the efficacy of lasmiditan.

  1. In the conclusion section, the author's description is rather general and lacks practicality. As a result, after reading the entire piece, one feels that the story ends abruptly, leaving the reader unsatisfied. It is suggested that the author provide more specific medication strategies for different patient subgroups (such as women and chronic migraine patients), or discuss how to balance efficacy and side effects.

Response: Thank you very much for your very important comment. We slightly shortened the conclusion sentence and added the following new suggestion for medication strategy (page 10, line: 5-9).

“We suggest medication strategies for the use of lasmiditan in East Asians. The initial dose should be 50mg and severity of side effects should be monitored. Patients who may benefit from lasmiditan include male, patients with CM, and patients receiving CGRP-mAb treatment.”

Round 2

Reviewer 1 Report

Comments and Suggestions for Authors

The reference to be quoted is: Singh, A., Gupta, D. & Sahoo, A.K. Acute Migraine: Can the New Drugs Clinically Outpace?.SN Compr. Clin. Med. 2, 1132–1138 (2020). https://doi.org/10.1007/s42399-020-00390-1

Author Response

Reviewer #1:

Thank you for providing detailed information about the literature.

Response: We cited your suggested paper as [28] and the following sentence was added in the Discussion (page 9, line: 37-41).

“Having discussed the efficacy, efficacy associated factors, and side effects of lasmiditan, it has been noted that the latest approved drugs for the management of acute migraine, such as lasmiditan, ubrogepant, and rimegepant, it remains difficult to justify their use except in special population/condition due to the lack of clinical trials investigating the consistency of long-term safety and efficacy [28].”

Reviewer 2 Report

Comments and Suggestions for Authors

The authors provided detailed responses and reasonable explanations to the comments raised by the reviewer and revised the manuscript accordingly. I think this paper can be accepted now.

Author Response

Reviewer #2:

Response: Thank you for taking the time to review the revised text. I am grateful that you deemed this paper acceptable.
